# Ancestral neuronal receptors are bacterial accessory toxins

Finaritra Raoelijaona[1,2], Joanna Szczepaniak[1,10], Adrien Schahl[3,10], James E. Bray[4,10], Jin Chuan Zhou[1,10], Lindsay Baker[1,2], Kamel El Omari[5], Edward Lowe[1,2], Yu Shang Low[6], Chandra M. Rodriguez[7], Michael J. Landsberg[6], J. Shaun Lott[7,8], Colin Kleanthous[1], Matthieu Chavent[3,9]✉, Martin CJ Maiden[4]✉ & Elena Seiradake[1,2]✉

Horizontal gene transfer events were crucial in the emergence of multicellular life. A striking example is the acquisition of Teneurins, putative surface-exposed toxins in bacteria that function as cell adhesion receptors in metazoan neuronal development. Here, we demonstrate the evolutionary relationships between metazoan and bacterial Teneurins. We use cryogenic electron microscopy and bioinformatic analysis to show that bacterial Teneurins harbour a toxic protein in a proteinaceous shell. They are rare but widely distributed across bacterial taxa and are predominantly seen in species with complex social behaviours, suggesting roles in cell-to-cell interaction. This work confirms that metazoan Teneurins are repurposed bacterial toxins that have evolved to be essential mediators of intercellular communication in all advanced nervous systems. Their acquisition was a key event in the evolution of metazoans.

Teneurins are a family of evolutionarily conserved proteins found across bilaterian metazoa and certain choanoflagellate species[1–5]. They are type II transmembrane receptors that evolved from an unknown bacterial precursor through horizontal gene transfer[2,3,6–10]. In mammals, Teneurins are prominently expressed in the central nervous system and are crucial for neuronal development[4,11–15]. In humans, Teneurin homologs have been implicated in diseases including sensory and motor dysfunctions, neurodevelopmental and psychiatric disorders, and cancers[16–23].

Teneurins consist of a small intracellular domain, a single transmembrane helix and a large extracellular region comprising eight epidermal growth factor-like repeats (EGF1-8), a cysteine-rich region, a transthyretin-like (TTR) domain, the characteristic Teneurin "superfold" and a C-terminal region containing an antibiotic-binding-like domain (ABD) and an HNH DNAse domain (Tox-GHH). The "superfold" is composed of a specialised fibronectin domain (FN-plug), a six-bladed NCL-1, HT2A, and Lin-41 (NHL) beta-propeller domain, and a large, anti-clockwise spiraling tyrosine-aspartate (YD) repeat domain shell[3,4,10,24–28] (Fig. 1a). Bioinformatic and structural studies of Teneurins suggest that the superfold shares homology with related bacterial rearrangement hotspot (RHS)/YD-repeat containing proteins[1,3,24–37]. These proteins are components of bacterial toxic effectors that facilitate host invasion, pathogenesis or alternatively compete or defend against predator species[32,38–41].

Phylogenetic analyses suggest Teneurin genes arose following the fusion of a prokaryotic proteinaceous toxin containing a

[1]Department of Biochemistry, University of Oxford, Oxford, UK. [2]Kavli Institute for NanoScience Discovery, University of Oxford, Oxford, UK. [3]IPBS, Université de Toulouse, CNRS, UPS, Toulouse, France. [4]Department of Biology, University of Oxford, Oxford, UK. [5]Diamond Light Source, Didcot, UK. [6]School of Chemistry and Molecular Biosciences, University of Queensland, St Lucia, QLD, Australia. [7]School of Biological Sciences, University of Auckland, Auckland, New Zealand. [8]Maurice Wilkins Centre for Molecular Biodiscovery, University of Auckland, Auckland, New Zealand. [9]Laboratoire de Microbiologie et de Génétique Moléculaire (LMGM), centre de biologie intégrative (CBI), Université de Toulouse, CNRS, UPS, Toulouse, France. [10]These authors contributed equally: Joanna Szczepaniak, Adrien Schahl, James E. Bray, Jin Chuan Zhou. ✉e-mail: matthieu.chavent@univ-tlse3.fr; martin.maiden@biology.ox.ac.uk; elena.seiradake@bioch.ox.ac.uk

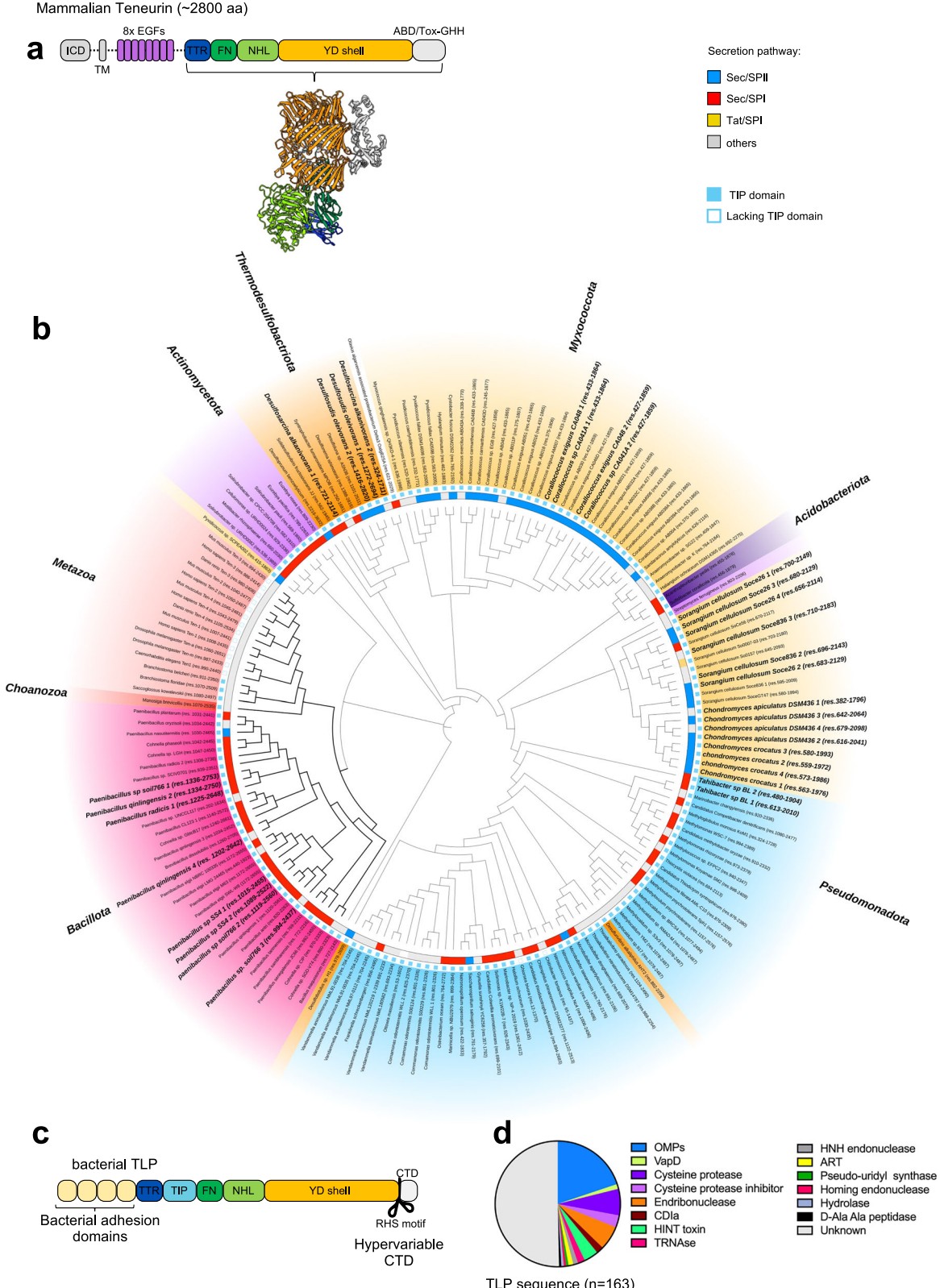

RHS/YD-repeat domain (the Teneurin 'superfold" and flanking domains) with a eukaryotic transmembrane protein gene[1,42,43]. Following structural characterisation of the vertebrate Teneurins encoded by these genes, a wide-spread, uncharacterized family of Teneurin-like proteins (TLPs) was discovered in bacteria[3,24]. As bacteria do not have a nervous system, bacterial TLPs must have functions beyond those currently known for the metazoan homologs. These important developments raised new

questions about the function(s) of bacterial TLPs and their evolutionary relationship to metazoan Teneurins.

Here, we show a small, but phylogenetically diverse group of bacteria encoding TLP genes in the PubMLST multispecies database. Structural analysis demonstrates that they possess Teneurin-like structural features while also revealing a highly variable C-terminal toxin module that is reminiscent of bacterial RHS/YD-repeat proteins[29].

**Fig. 1 | TLPs are found across the bacterial kingdom. a** Teneurin domain organisation. ICD (intracellular domain), TM (transmembrane), EGF (Epidermal growth factor), TTR (transthyretin-like), FN (Fibronectin type 3), NHL (NCL-1, HT2A, and Lin-41 domain), YD-shell (Tyrosine-Aspartate repeat domain), ABD (Antibiotic binding domain), Tox-GHH (Toxin-like HNH DNAse domain). The cartoon representation of the Teneurin extracellular domain was based on the chicken Teneurin 2 (PDB id: 6FB3[24]). **b** Maximum likelihood phylogenetic tree based on an alignment of the Teneurin/TLP superfold. Species names are color-coded according to the phylum: Bacillota (magenta), Actinomycetota (purple), Pseudomonadota (blue), Myxococcota (yellow), Thermodesulfobacteriota (orange), Acidobacteriota (dark purple), Choanozoa (red), Metazoa (light red). Bacterial genomes that encode

multiple TLP genes are highlighted in bold. The presence of a TIP (Teneurin insertion protein) domain is indicated by a solid blue square. Sequences with a predicted signal peptide are colored according to the predicted trafficking machinery: SEC translocon SEC/SPI (red), SEC/SPII (blue), TAT translocon TAT/SPI (yellow), other (light gray). **c** Bacterial TLP domain organisation, colours in analogy to (**a**). Unlike Teneurins, bacterial TLPs lack an intracellular domain and contain additional variable adhesion domains. The TIP domain is inserted between the TTR and FN-plug domains. The C-terminal (CTD) domain is highly variable. **d** Predicted enzymatic functions of TLP C-termini. We used AlphaFold[49] and structural homology searches to infer functionality.

We also confirm the presence of associated immunity genes, suggesting that bacterial TLPs are a distinct subtype of bacterial polymorphic toxins[30]. Our data show that bacterial TLPs are an accessory platform for the delivery of toxins in bacterial competition systems, a biological role that has been lost in metazoans where instead they mediate essential cell adhesion and signaling functions.

## Results

### Bacterial TLPs are found in a small group of species, yet widespread across the bacterial kingdom

We performed a comprehensive homology search to identify TLP genes in the PubMLST multispecies database (pubmlst.org/species-id)[44], using the profile Hidden Markov Models (HMM) constructed for each of the three domains which constitute the Teneurin superfold (FN-plug, NHL, YD-shell). The search returned 139 TLP positive genomes (Supplementary data 1 and 2) spread across Gram-negative and Gram-positive bacterial groups in six phyla: Pseudomonadota, Myxococcota, Thermodesulfobacteriota, Acidobacteriota, Bacillota, and Actinomycetota (Fig. 1b, Supplementary Fig. 1a). Very few species within each phylum (less than 3%) encode a TLP gene, apart from Myxococcota, of which >20% encode at least one TLP gene (Supplementary Fig. 1b). Our phylogenetic analysis suggests that TLP sequences within the same phyla are generally conserved. Most bacterial species encode one copy of the TLP gene. However, some genomes carry up to four copies, as do mammalian genomes[8]. At least some of these additional copies have likely been acquired by independent gene transfer, rather than gene duplication, as they are most closely related to TLPs found in other species from the same family (Fig. 1b). The eukaryotic Teneurins cluster most closely with TLPs from the *Paenibacillaceae* family, which form a distinct clade (Fig. 1b).

Analysis of these TLP sequences (Supplementary data 3) shows that the superfold domain organisation remains invariant across phyla, while the flanking N-terminal and C-terminal regions are variable. Diverse adhesion domains serve as N-terminal modules (Supplementary Fig. 1c) that presumably govern how TLPs are displayed on the cell surface. We found that 96% of the TLP N-termini contain a Teneurin insertion protein (TIP) domain, which is absent in the metazoan Teneurins (Fig. 1a, b). Based on putative signal sequences, the Gram-negative TLPs, such as those within the Myxococcota phylum, are predicted to be targeted to the SEC/SPII secretion system. These TLPs also contain predicted lipoprotein domains that could facilitate attachment to the membrane surface. In contrast, putative signal sequences found in TLPs from Gram-positive bacteria are predicted to use the SEC/SPI secretion system and contain predicted S-layer or cell wall binding domains that would anchor them to the cell surface (Fig. 1b, Supplementary Fig. 1c)[45,46]. Protein translocation in bacteria is mediated by a variety of secretion systems[47,48], and the absence of a predicted signal peptide in some TLP sequences suggests they may utilize alternative mechanisms for secretion.

Following on from the superfold, the C-terminal domain (CTD) is typically much smaller than the N-terminal region, averaging ~15 kDa, and is highly polymorphic (Supplementary Fig. 1). The hypervariability within the different CTD sequences, and lack of functional annotation,

led us to perform structural predictions using AlphaFold[49]. We searched for structural homologs using the DALI and Foldseek programs[50,51] to derive structural homology information for those models that were predicted with high confidence scores. This analysis revealed structural similarities to small proteases, inhibitors, peptidases, hydrolases, Hedgehog/INTein (HINT) toxins, nucleases, pseudouridine synthases, ADP-ribosyl transferases, and outer membrane proteins (OMPs) (Fig. 1d). The wide variety of enzymatic functions associated with these domains, and the similarity to toxin proteins known to be utilized in bacterial competition or host interaction suggest that these domains may function as cytotoxic effector molecules. Our results suggest that TLP genes are present in a small yet widely distributed group of bacteria, representing a distinct class of polymorphic toxin systems. Moreover, TLP genes are particularly associated with bacteria that exhibit complex social behaviors suggesting that they may contribute to intercellular cooperation.

### The bacterial TLP structure is homologous to its metazoan counterparts

Previously, we identified a TLP gene[3,24] within the genome of *Bacillus subtilis* CW14 (now re-named *Bacillus inaquosorum*[52,53]) (Fig. 2a). To provide insight into bacterial TLP molecular architecture, we expressed, purified and determined cryo-EM structures of the protein encoded by the gene (*Bi*TLP). Using a construct that lacks the predicted N-terminal bacterial adhesion domains (residues 1-397) (Supplementary Table 1, Supplementary Fig. 2a–e, Supplementary Fig. 3a, b) we determined a map with an average resolution of 2.1 Å. We also determined the structure using a full length *Bi*TLP construct (*Bi*TLP^FL, Supplementary Table 1, Supplementary Fig. 4a–e), which has a lower average resolution. As the additional adhesin domains were not resolved in maps of *Bi*TLP^FL, the analysis here focuses on the higher resolution map. The *Bi*TLP model was built de novo by modifying a partially correct model generated with AlphaFold2 (Supplementary Fig. 2f)[49]. The resulting model reveals a similar organization to the metazoan homologs, with the RHS/YD-shell consisting of three layers of antiparallel beta sheets rotating anticlockwise along the central axis (Fig. 2b, c). The N-terminal side of the shell is sealed with the FN-plug, and the NHL domain is positioned at an angle of approximately 60° from the axis of the shell (Fig. 2c). A closer view of the map shows that the map density stops abruptly following residue L2144 (Supplementary Fig. 3c). Sequence analysis of this region suggests the presence of a PxxxxDPxG motif which is characteristic of the conserved RHS core domain cleavage site (Fig. 2a)[54]. In analogy to the bipartite DPxG-X$_{18}$-DPxG motif found in other RHS proteins, where the two aspartic acids both play a role in catalysis, we found two DPxG motifs twenty residues apart from each other. In some species, the glycine residue for the first motif is substituted by a leucine or an arginine (Supplementary Fig. 5a). The structure of the catalytic aspartic acid motif is conserved and is located proximal to a structurally conserved arginine which has previously been shown as essential for catalysis[29] (Fig. 2d, Supplementary Fig. 5b). Indeed, mutation of this arginine reduces autoproteolytic activity (supplementary Fig. 5c), consistent with what had been established previously for the RHS/YD-repeat-containing BC subunit of

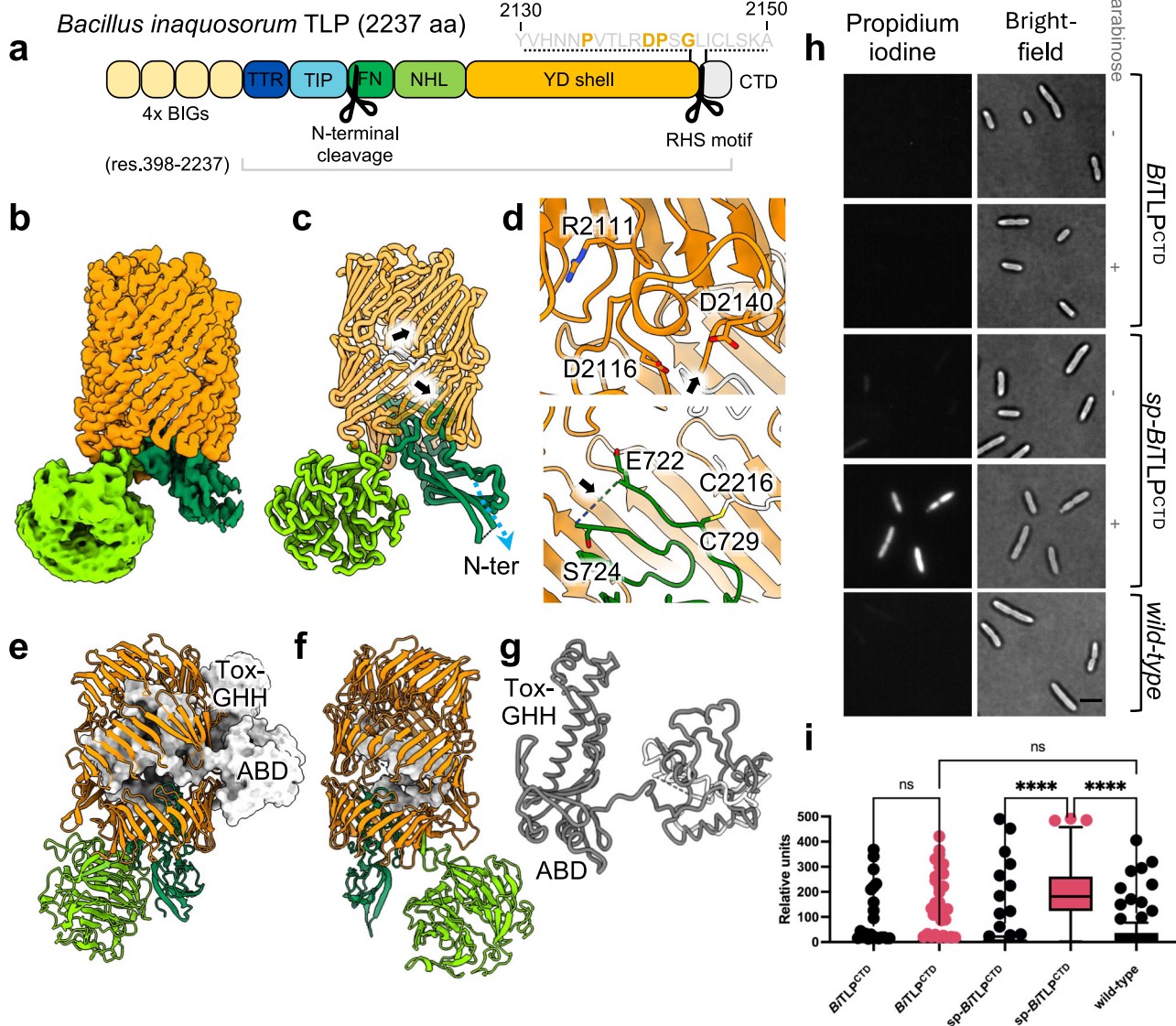

**Fig. 2 | The *Bacillus inaquosorum* TLP structure is homologous to mammalian Teneurin. a** Schematic representation of *Bi*TLP domain organisation. Bacterial Immunoglobulin-like domain, BIG (yellow), TTR (dark blue), TIP (aqua blue), FN-plug (forest green), NHL (lime green), YD-shell (orange), CTD (light grey). Putative autoproteolytic cleavage sites are indicated with scissors. The RHS motif is highlighted (yellow letters). The construct used for structural studies (residues 398-2237) is outlined by a grey line. **b** the cryogenic electron microscopy density map is colored according to the domain it represents, FN-plug (forest green), NHL (lime green), YD-shell (orange), CTD (light grey). **c** Cartoon representation of the *Bi*TLP model, colored according to (**b**). Black arrows indicate autoproteolytic cleavage sites. The N-terminal chain end is indicated with a blue arrow. **d** Zoomed views. Top: the cleavage site within the RHS motif, containing the aspartyl protease catalytic residues D2116 and D2140 and the conserved R2103. Bottom: the cleavage site in the FN-plug domain is highlighted by a black arrow. The residues flanking the

cleavage site are indicated (E722 and S724). C719 forms a covalent bond with C2216, linking the N-terminal fragment to the C-terminal fragment. **e** Cartoon representation of chicken Teneurin 2 (PDB id:6FB3[24]), colored in analogy to the *Bi*TLP domain organisation in (**b**). The C-terminal region downstream of the YD-shell is shown as a surface representation (light grey). **f** As (**e**), but showing *Bi*TLP, in the same cartoon/surface representation and equivalent colour scheme. **g** Overlay of the C-terminal regions of chicken Teneurin 2 (dark grey) and *Bi*TLP (light grey). **h** Propidium accumulation in *E.coli* cells expressing the *Bi*TLP CTD with or without an N-terminal secretion signal peptide were imaged using HiLo microscopy. Scale bar =5um. **i** Quantification of propidium accumulation in *E.coli* cells, presented as box plot showing the median (central line), the first to the third quartile (box limit) and the minima and maxima (whisker). Data represents an average of 3 independent repeats (*n* = 3). Two-way ANOVA with Tukey's multiple comparisons test was performed; adjusted p-value ****<0.0001, ns=non-significant.

bacterial ABC toxins[29]. In mammalian Teneurins the motif is absent (Supplementary Fig. 3d). An additional chain break was observed within the FN-plug domain, in a loop located deep within the YD-shell domain, between residues E722 and S724 (Fig. 2d, Supplementary Fig. 3b, e). In the *Bi*TLP^FL structure, this chain break could be unequivocally identified as occurring between G723 and S724 (Supplementary Fig. 5d) as also confirmed by N-terminal sequencing (Supplementary data 4). Sequence analysis of this region suggests that S724 is highly conserved among TLP family members (Supplementary Fig. 3f). Autocatalytic cleavage upstream of the shell domain has been

observed in other RHS/YD-repeat-containing toxins where cleavage is important for the release of the toxin[34,35,37,55,56]. Close inspection of the cryo-EM map density indicates a small extra density which could indicate the presence of a metal ion in this area (Supplementary Fig. 5e).

Despite their presence during sample preparation, densities for the domains located upstream of the FN-plug were not observed in our maps (Fig. 2b) and they were therefore excluded from the final model. However, we observed substantial additional density (~6640 Å³) within the YD-shell (Supplementary Fig. 3g). Tracing this density revealed part

of the CTD. Of the 92 amino acids (residues 2145-2237) in this domain, all but the regions 2166-2171, 2190-2194 and 2219-2237 are accounted for in our model (Supplementary Fig. 3a, g). Most of the traced CTD nestles closely to the inner surface of the YD-shell. Surprisingly, C2216 forms a disulfide bond with C719, covalently linking it to the FN-plug (Fig. 2d). In contrast to the published metazoan Teneurin structures, where an uncleaved C-terminus leads through an opening in the YD-shell to form the ABD and Tox-GHH domains outside the shell (Fig. 2e)[4,10,24–28], there is no evidence of density outside the YD-shell and no opening in the YD-shell through which the C-terminus could exit (Fig. 2b, f). Indeed, the *Bi*TLP CTD shares no structural or sequence homology with the Teneurin CTD (Fig. 2e–g). Bacterial TLP CTDs are relatively short in sequence (Supplementary data 5), averaging around 15 kDa. This size constraint may represent an evolutionary limitation imposed by the RHS/YD-shell inner cavity, which has a volume of approximately 44,000 Å³ in *Bi*TLP. Taken together, we conclude that the CTD, while cleaved at the conserved RHS core domain cleavage site, remains entirely hidden by the YD-shell in the *Bi*TLP example. It may remain associated with the N-terminal domains following release from the shell via the C2216-C719 disulfide.

## The C-termini of TLPs harbour diverse toxic functions

The presence of autoproteolytic sites in *Bi*TLP are reminiscent of those seen in other bacterial RHS and YD-repeat proteins which typically function as toxins[29,35–37,40,55–58]. In particular, the presence of two cleavage sites and the possibility that the N- and C-terminal fragments liberated by these events may remain associated following proteolysis suggests an evolutionary connection to RHS effectors that function as toxins in concert with bacterial type VI secretion systems. These toxins also encapsulate cleaved CTDs, which represent a variety of cytotoxic enzymes[59–61]. We therefore investigated whether *Bi*TLP harbours a toxin in its C-terminal region. Sequence analysis of the *Bi*TLP CTD fragment (*Bi*TLP^CTD) suggests that the fragment is rich in hydrophobic residues, with a predicted transmembrane region between residues 2165 and 2185 (Supplementary Fig. 3a). We used Coarse-Grain Molecular Dynamics (MD) to simulate the *Bi*TLP^CTD in the presence of a model membrane composed of 25% phosphatidylglycerol and 75% phosphatidylethanolamine, in analogy to the *E. coli* inner membrane. 10 independent simulations of 3 μs duration showed rapid association of *Bi*TLP^CTD with the membrane (Supplementary Fig. 6a). Simulation using multiple copies of *Bi*TLP^CTD leads to membrane deformation (Supplementary Fig. 6b).

We expressed *Bi*TLP^CTD with an N-terminal secretion signal peptide (sp-*Bi*TLP^CTD) in *E. coli* and found that expression of this protein inhibited cell culture growth (Supplementary Fig. 6c). Expression of *Bi*TLP^CTD without a signal peptide did not inhibit growth (Supplementary Fig. 6c). The observation of membrane deformation in our MD simulation encouraged us to investigate whether the CTDs affect membrane integrity. We treated cells expressing either *Bi*TLP^CTD or sp-*Bi*TLP^CTD with propidium iodide stain which does not penetrate intact cell membranes. Cells expressing sp-*Bi*TLP^CTD were permeable to the stain (Fig. 2h, i). These results, and the diverse enzymatic functions we predicted for different TLP CTDs led us to hypothesize that bacterial TLPs could utilize a variety of mechanisms to modify their targets (Fig. 3a). To investigate this further, we selected nine representative TLP CTDs from different organisms, for which structures could be predicted and functions consequently inferred (Supplementary Fig. 6d). Despite having no knowledge of the physiological targets for these putative toxins, we observed that three of the chosen CTDs inhibited *E. coli* cell growth in our liquid cultures as well as on soft agar (Fig. 3b). We focused on the CTD of *Methylosarcina fibrata* TLP (*Mf*TLP^CTD) for which AlphaFold confidently predicted an ADP-ribosyl transferase (ART) fold (Fig. 3c). ART proteins convert NAD^+ to nicotinamide to transfer ADP-ribosyl groups onto target proteins and can act as virulence factors, *e.g.* in diphtheria and cholera toxins[62,63]. The ARTs in these toxins depend on catalytic HYE and RSE motifs in

their active sites, respectively[64]. *Mf*TLP^CTD contains a HSE motif (H2319, S2364, E2405), which could constitute the catalytic triad (Fig. 3c). In agreement with this hypothesis, mutation of the conserved glutamate residue (E2405) to alanine restored cell growth in *E. coli* (Fig. 3d). As expected for an ART-dependent toxin, expression of wild-type *Mf*TLP^CTD, but not the mutant, led to depletion of cellular NAD^+ (Fig. 3e). In agreement with a different mechanism for growth inhibition, the sp-*Bi*TLP^CTD did not deplete NAD^+. Taken together, these results demonstrate that different TLP CTDs encased within the RHS/YD-shells are cytotoxic to bacteria through different mechanisms.

## Bacterial genomes encode TLP-associated immunity proteins

Given that the superfold is conserved across bacterial species, we hypothesized that it could perhaps confer host immunity to the C-terminal toxin by trapping it inside the RHS/YD-shell. To test this hypothesis, we engineered chimeric constructs consisting of the *Bi*TLP superfold fused to the CTD from *M. fibrata*, *Acanthopleuribacter pedis* or *Methylocaldum sp*. We found that *E. coli* cells were sensitive to the expression of these chimeric constructs (Supplementary Fig. 7a), in disagreement with the idea that the RHS/YD-shell confers a protective function for the host cell, at least in these experiments (Supplementary Fig. 7a). Further analysis of TLP-encoding operons revealed that an additional gene is found immediately downstream of almost all TLP open reading frames (Fig. 4a, b, Supplementary Fig. 7b, Supplementary Data 6), reminiscent of the arrangement seen in toxin-antitoxin systems[60]. This led us to hypothesize that these highly diverse genes could act as specific immunity proteins to protect the TLP expressing cell from the C-terminal toxin by directly binding to it, as seen in other bacterial polymorphic toxin systems[60]. Co-immunoprecipitation experiments supported this conclusion, showing that the *M. fibrata* immunity protein associates with the *Mf*TLP^CTD (Fig. 4c). Co-expression of this putative immunity protein in the NAD^+ depletion assay confirms that the ART activity of *Mf*TLP^CTD is inhibited by its presence (Fig. 4d). Co-expression also restored *E. coli* growth on soft agar (Fig. 4e). Structural predictions using AlphaFold2[49] further supports a direct interaction between these 'immunity proteins' and the corresponding TLP C-termini found in different species (Fig. 4b, Supplementary Fig. 7c). We therefore conclude that different bacterial TLPs are co-expressed with matching immunity proteins encoded in the same operon.

## Discussion

Teneurins are best understood in mammalian nervous systems, where they function as cell signalling receptors. However, less is known about their prokaryotic ancestry. Here we show that their genes emerged from a bacterial precursor that functioned as a toxin prior to the evolution of nervous systems[1,42,43,65] (Fig. 5a). Bacteria commonly live in highly dense communities where cell-to-cell interactions are essential for survival and adaptation. Consequently, they have developed numerous strategies to either cooperate with, or compete against each other, including the evolution of toxin systems that respond to cell-cell interactions. The results we presented here for TLPs are characteristic of polymorphic toxins[30,60,66], which are one of the most complex and dominant bacterial conflict systems[67]. We found TLPs particularly highly represented in *Myxococcaceae* of which many such as *Corallococcus* display a predatory lifestyle, killing and consuming a wide range of prey through the secretion of antimicrobial substances[68,69]. It is therefore likely that TLPs function in bacterial warfare, where immunity genes are crucial for kin and non-kin recognition (Fig. 5b). Of the polymorphic toxins, TLPs share structural similarities with RHS/YD proteins. These are often associated with the type VI secretion system and deployed by Gram negative species to kill or inhibit the growth of neighbouring cells[35–39,55,56,58]. In another example, the tripartite ABC toxin complex requires a type 10 secretion system and targets insect host cells[57,70,71]. Like the bacterial RHS/YD toxins, TLPs package their

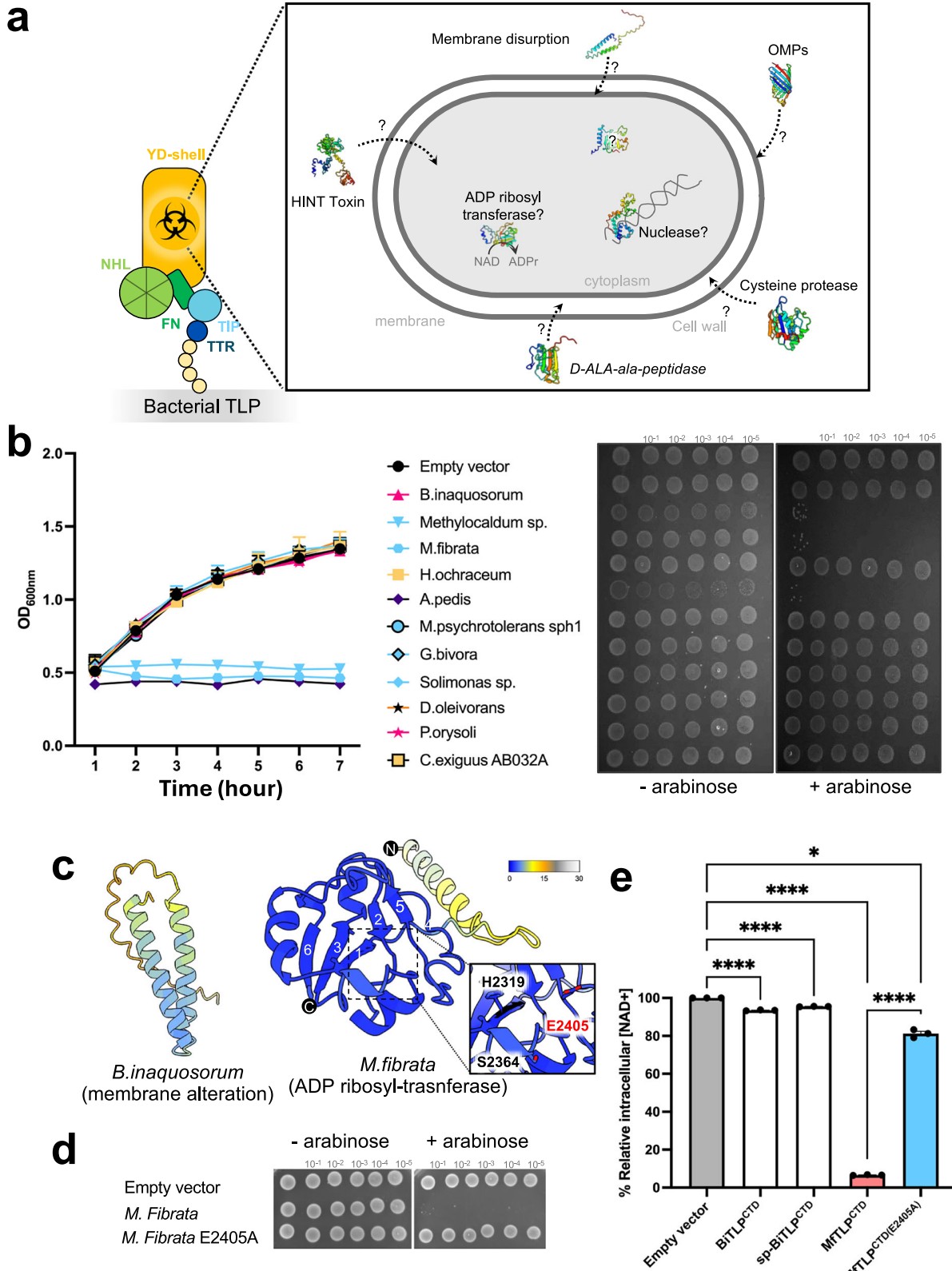

functional toxins within a protein shell. However, in the absence of any obvious delivery machinery, the mechanism via which they are delivered to target cells and released from the shell remains to be elucidated. In addition to their toxin function, RHS/YD proteins play roles in the social behaviour of bacteria, such as in S motility[72]. In analogy, we found TLP positive genomes predominantly in *Paenibacillaceae* and *Myxococcaceae*, families that exhibit complex social behaviours[73,74].

This leaves open the possibility that bacterial TLPs function in cell-to-cell communication within microbial communities.

Our phylogenetic analysis was based on the superfold sequence and suggests that *Paenibacillaceae* TLPs are the closest known relatives of eukaryotic Teneurins. In contrast to Teneurins which contain a C-terminal DNase fold, *Paenibacillaceae* TLPs contain a predicted outer membrane protein in their CTDs. It is possible that the

**Fig. 3 | Bacterial TLPs harbour diverse enzymatic functions. a** Schematic representation of bacterial TLP, where the YD-shell carries the putative 'toxic' CTD associated with different functions. Representative CTDs, selected for toxicity assays, are shown with their predicted functions: OMP (Outer membrane proteins): *Paenibacillus oryzisoli*, *Desulfosudis oleivorans*; ART (ADP ribosyl transferase): *M. fibrata*; HINT toxin: *Hyalangium ochraceum*, *Acanthopleuribacter pedis*; nuclease: *Methylocaldum sp.*, *Methylovolum psychrotolerans* sph1; protease: *Solimonas sp.*, *Corallococcus exiguus* AB032A; peptidase: *Ghiorsea bivora*. **b** Toxicity assay in *E. coli* cells expressing different bacterial TLP CTD as shown in (**a**). Left: Growth in liquid LB medium of *E. coli* top10 cells carrying a pBAD vector control (empty vector) or plasmids directing the expression of each CTD. Optical density at 600 nm was measured every hour for 6 hours following induction with 2% L-Arabinose. Points show mean ± SEM, *n* = 3 replicates. Curves are colored according to the corresponding phylum: Bacillota (magenta), Pseudomonadota (blue), Myxococcota (yellow), Thermodesulfobateriota (orange) and Acidobacteriota (dark purple).

Right: Bacterial growth of serially diluted cells on soft agar after overnight incubation at 37 °C. To induce gene expression, 2% L-arabinose was added to the medium. **c** Predicted AlphaFold models of *B.inaquosorum* and *M.fibrata* CTDs. Each model is color-coded based on the AlphaFold confidence score. A closer view of the *M.fibrata* catalytic triad (H2319, S2364, E2405) responsible for NADase activity is indicated, with the conserved glutamate residue shown in red. The strands are numbered according to the conventional ART fold annotation. **d** Toxicity assay in *E. coli* Top10 expressing either the *M. fibrata* CTD or the catalytically inactive mutant (E2405A) on soft agar as in panel **b**. **e** Intracellular NAD+ levels measured in *E. coli* Top10 cells expressing the TLP CTDs of *B.inaquosorum* or *M. fibrata* (wild type or the inactive mutant E2405A). NAD+ levels were measured 1 h post-induction. Luminescence signal was recorded for 2 h and averaged over 3 repeats. Data represent a relative percentage normalised to the empty vector. Bars show mean ± SD, *n* = 3 independent experiments. One-way ANOVA with Tukey's multiple comparisons test was performed; adjusted p-values ****<0.0001, *<0.01.

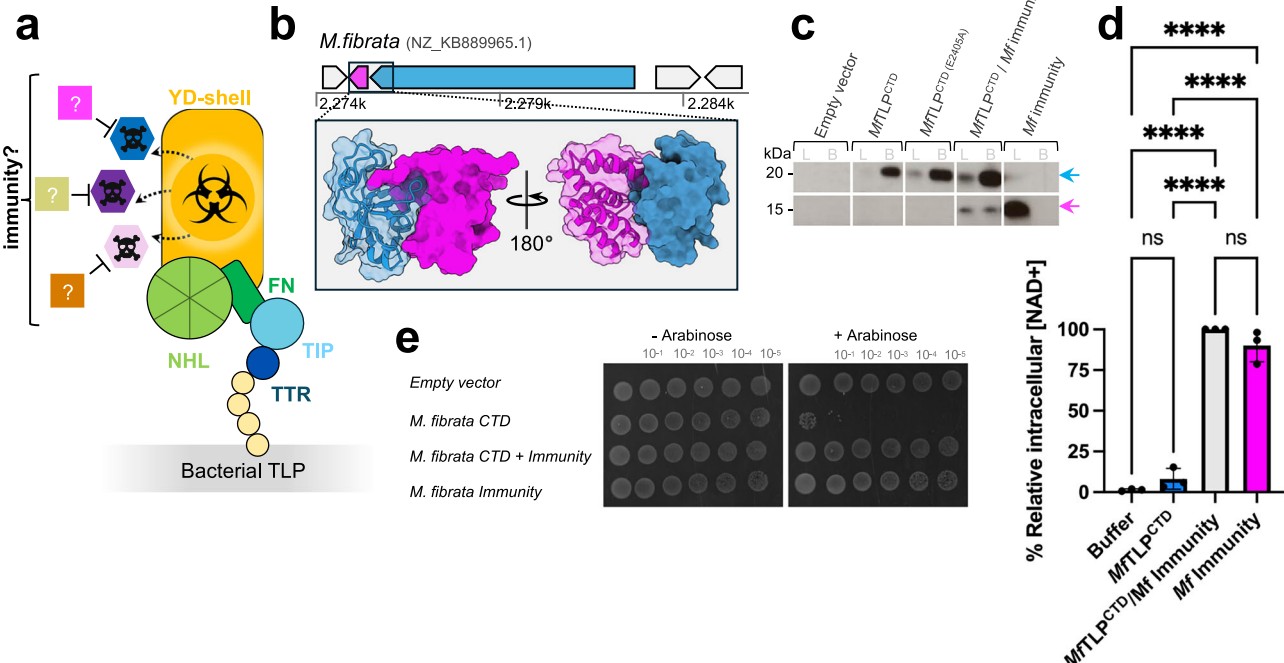

**Fig. 4 | Bacterial TLPs are encoded as an effector/immunity pair. a** Illustration of the presence of a matching immunity gene that neutralises the CTD 'toxic' activity. **b** Top: Analysis of the *M.fibrata* genome reveals a second, small gene in the genomic neighbourhood of TLP. The TLP gene (blue) is encoded on the reverse strand, followed by the cognate immunity (magenta). Surrounding genes are colored grey. Bottom: Predicted AlphaFold model of *M.fibrata* CTD/immunity heterodimer shown as ribbon and/or surface. **c** Immunoblot following pull down assay confirms the predicted interaction shown in panel b. *M.fibrata* CTD was co-expressed with its immunity gene in *E. coli* Top10. The CTD was fused with an N-terminal twinstrep-HA tag to allow immobilisation on streptavidin-agarose beads, while the immunity protein, tagged with a Flag tag, served as the prey. Both protein bands were

detected using anti-HA and anti-Flag antibodies. **d** Intracellular NAD+ levels measured in *E. coli* Top10 cells expressing *M.fibrata* CTD, the immunity gene or both. NAD+ levels were measured 1 h post-induction. Luminescence signal was recorded for 2 h and averaged over 3 repeats. Data represent a relative percentage normalised to the CTD/Immunity complex. Bars shown mean ± SD, *n* = 3 independent experiments. One-way ANOVA with Tukey's multiple comparisons test was performed; adjusted p-value ****<0.0001, ns=non-significant. **e** Co-expression of the matching immunity gene restores *E. coli* growth. In a toxicity assay using *E. coli* Top10, we expressed *M. fibrata* constructs described above (**d**). Bacterial cell cultures were serially diluted on soft agar and incubated overnight at 37 °C, with protein expression induced by the addition of arabinose.

acquisition of the Teneurin superfold and the CTD were evolutionarily decoupled[24,54], or that the original prokaryotic ancestor is now extinct. The presence of a Teneurin gene in choanoflagellate species suggests that it was acquired early in metazoan evolution[65,75]. Choanoflagellates are free-living organisms that prey on bacteria as a food source[76,77]. Therefore, eukaryotic Teneurins likely arose from an independent horizontal gene transfer as a consequence of this predatory relationship[76,78]. Whether the gene uptake event was driven by their function as toxins, *e.g* to kill bacterial prey, or as cell adhesion molecules, *e.g* facilitating interaction with bacterial cells with the aim of capturing them[78], is yet to be investigated.

There is evidence that isolated metazoan Teneurin CTDs could possess nuclease activity and regulate cell apoptosis[31]. However, Teneurins are best known for their functions as cell surface signaling receptors that mediate crucial cell-to-cell communication via homophilic[10,27,79] and heterophilic interactions, such as with the adhesion G-protein-coupled receptor Latrophilin[4,15,26,80] (Fig. 5c). Moreover, homodimerization of some RHS/YD proteins, *e.g* RhsP and RhsA found respectively in *Vibrio* and *Pseudomonas* species, has been reported[35,36] although the functional significance of dimerization in these systems remains to be explored. While the evidence we present here for cell surface localization of TLPs is limited to functional homologies

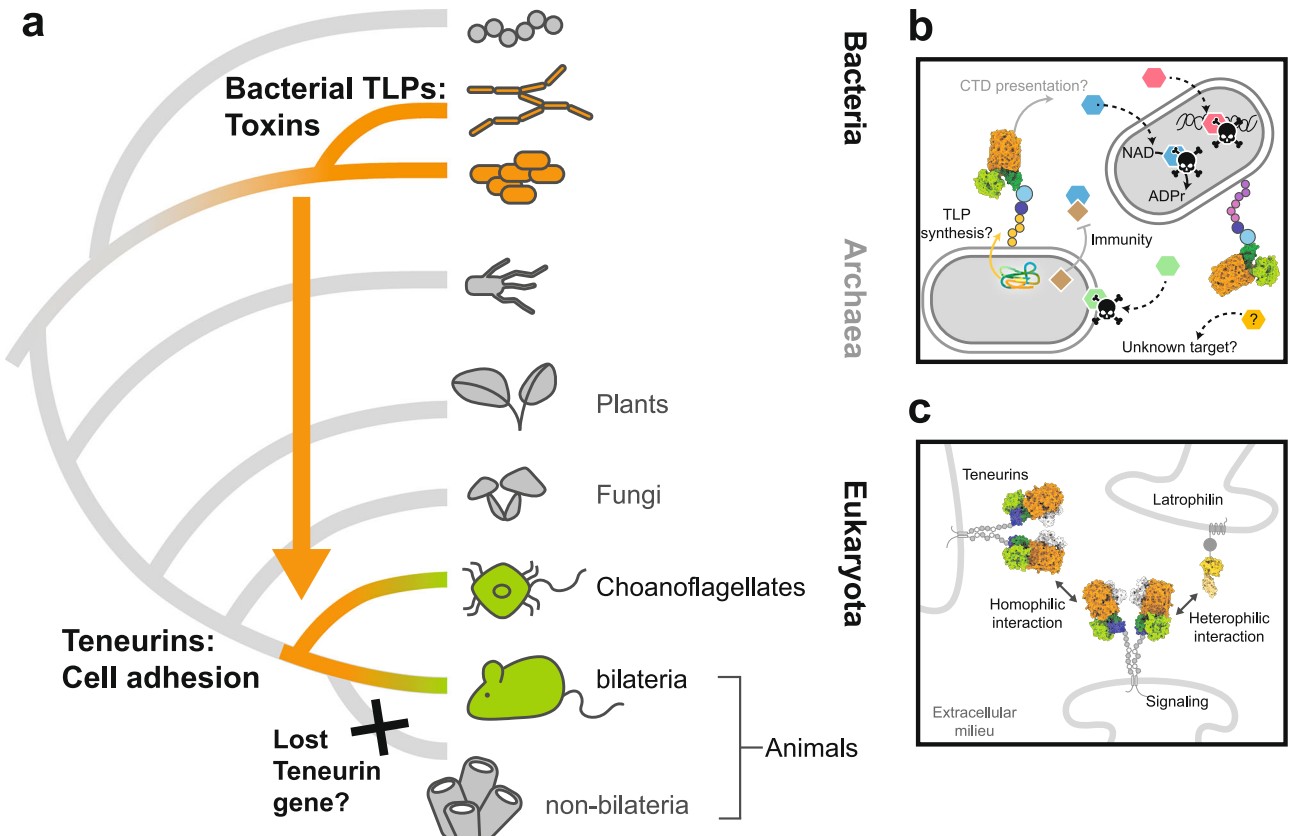

**Fig. 5 | Teneurins emerge from a bacterial toxin precursor through horizontal gene transfer. a** Schematic tree depicting Teneurin gene transfer event (orange arrow) from an unknown bacterial precursor to the choanoflagellates-animals clade. The ancestral gene 'bacterial TLP' may function as toxin but later co-opted to cell adhesion in bilaterian animals. Non bilaterian animals refer to sponges, placozoa, ctenophora and cnidaria. Black cross represents the absence of Teneurins gene. **b** Bacterial TLP family hypothetical mechanism of action. Upon synthesis, TLPs are possibly exposed on the cell surface, enabling CTD trafficking. Released CTD, triggered by a yet to be discovered mechanism are represented as hexagonal shape, with colours representing the various enzymatic functions. Dashed arrows indicate potential action sites within another bacterial cell or on an unknown target. Solid squares represent immunity protein. **c** In analogy to (**b**), Bilaterian Teneurins mediate cell to cell adhesion in a receptor-ligand interaction. They bind other Teneurin molecules in a homophilic interaction and/or the adhesion G protein-coupled receptor Latrophilin in a heterophilic manner (PDB id:6SKA[4]). Bilaterian Teneurins also have an intracellular region involved in signalling.

inferred from bioinformatic sequence analyses, localization of bacterial TLPs to the cell surface may also allow them to engage in cell-to-cell contact and signalling functions. Eukaryotic tissues, such as the nervous system, and bacterial quorum sensing share some similarities in how cell-cell-communication is achieved[81]. In biofilm communities, bacteria communicate through ion-based signals which resemble electrical signals used by neurons[82]. While such communication mechanisms may be redundant or accessory for bacteria, in agreement with the widespread but low incidence of TLP genes in our database, co-opting these receptors must have been essential for the evolution of animals, where Teneurins are conserved in all organisms with a centralized nervous system. Taken together, this suggests that the acquisition of the Teneurin gene represents an important event in the evolution of complex multicellular life.

## Methods

### Teneurin-like protein identification using hidden Markov model searches

The sequence region of protein WP_088111228.1 corresponding with the Teneurin-like protein (TLP) superfold (residues 673 to 2139) was extracted from the NCBI Protein database (https://www.ncbi.nlm.nih.gov/protein) and searched against the non-redundant protein sequence database (NR) using the NCBI BLAST website (16th Dec 2022).

The top 90 matching sequence regions with at least 36% sequence identity and 99% query coverage were extracted and aligned using MAFFT (v7.490[83]). The resulting multiple sequence alignment was split into 3 separate alignments corresponding with the 3 TLP superfold domains: FN-plug (region 673-928), NHL (929-1255) and YD shell (1256-2139). A single Hidden Markov Model (HMM) was constructed for each domain using hmmbuild (HMMer software package, v3.1b1, http://hmmer.org) and the three models were combined into a single HMM library.

A dataset of 481,806 bacterial genomes, comprising 16,838 different species, was identified as present in the PubMLST Multi-species database[44] on 18th January 2023 (https://pubmlst.org/species-id). Each genome was downloaded from the database, subjected to a six-frame translation (EMBOSS transeq, v6.6.0.0) and the protein sequences were scanned against the HMM library using hmmscan (HMMer software[84]). All operations were performed on a Dell PowerEdge R815 Server with 512 Gb of RAM and 64 CPU cores.

Sequence matches with an E-value of 1e-10 or less and at least 30% domain overlap were shortlisted for manual inspection of the results. In total 143 genomes from 93 different species were identified as containing the 3 TLP superfold domains, adjacent to each other on the same reading frame and in the correct domain order.

### Sequence conservation and phylogenetic tree inference

Each TLP sequence was retrieved from the homology search mentioned above. A multiple sequence alignment (MSA) of all bacterial TLP full length sequences and chosen metazoan representative was

generated using MUSCLE program of the jalview package[85,86]. Sequences corresponding to *Chitiniphilus eburneus*, *Olavius algarvensis* associated proteobacterium Delta3 (OalgA1CA) and *Mizugakiibacter sediminis* were removed from final analysis given that the sequence were either partial or do not correctly align with the other TLP sequences.

The conserved region corresponding to superfold sequences including the FN-plug, NHL and the YD-shell were isolated and further cleaned using TrimAL with default preset from the NG phylogeny.fr online server[87–90]. The cleaned output from TrimAL was the used as a query for tree inference using IQ-TREE server with default preset and an ultrafast bootstrap of 1000×[90]. The maximum likelihood consensus tree was then uploaded on Itol server for visualization and annotation[91].

### Bacterial TLP N-terminal domain annotation and secretion pathway prediction
Full length protein sequences were subjected to signal peptide prediction using the signal P5 server[92]. The CD search and interporscan server were used to search form domain homology and annotate domain function[93–95].

### Structure prediction and comparison with available structures on the PDB database
Structure predictions were performed using Alphafold2 in a local version of ColabFold[49,96]. A complete MSA sampling using the default database was performed along 3 cycles, generating 5 structures corresponding to each bacterial TLP CTD sequence. The CTD corresponds to the region downstream of the RHS cleavage site 'DPxG' motif. Among each set of structures, the one presenting the highest pLDDT score was chosen for structural homology and functional inference using DALI or foldseek server[50,51].

### Vector and cloning
All cDNA used for the project was commercially synthesised using Genscript. *Bacillus inaquosorum* constructs (residues 398-2237, *Bi*TLP) and R2103A (*Bi*TLP^R2103A) were subcloned into a petMCN vector carrying a N-terminal His tag and a TEV protease site[97]. In addition, *Bi*TLP and *Bi*TLP^R2103A carry a N-terminal StrepII tag, a central FLAG tag (within the RHS associated core) and C-terminal HA tag. All bacterial TLPs CTD were cloned into a pBAD vector fused with a N-terminal twin-strep-HA tag[98]. *Bi*TLP^FL (residues 1-2237) was commercially synthesised and cloned by Azenta into a pProExHta vector containing an N-terminal His tag. A list of all plasmids and primers used in this study is provided in Supplementary Table 2.

### Protein expression and purification
The *Bi*TLP construct was transformed into *E.coli* BL21. Bacterial cells carrying the vector were selected for growth in TB media supplemented with 100 µg/µl ampicillin and protein expression was induced with 0.5 mM IPTG at 18 °C for 14 to 16 h. Bacterial pellet were harvested by centrifugation for 30 min at 4000 rpm at 4 °C. Bacterial cells were resuspended in lysis buffer (50 mM Tris pH 7.5, 300 mM NaCl, 5 mM imidazole, 0.1%Tritonx100, 5 mM mercaptoethanol). Cell lysis was performed by sonication (amplitude: 45, 30 s x 5). Crude cells were then clarified by centrifugation for 1 h at 4 °C, 30000 $g$. Supernatant was loaded onto a prepacked His-Trap column (GE). *Bi*TLP protein was eluted with wash buffer containing 500 mM imidazole. Protein was incubated overnight with TEV protease at 4 °C. Following treatment, the sample was reloaded onto a pre-equilibrated His-Trap column (GE). The flowthrough containing the cleaved sample were concentrated and loaded onto a Superose6 10/300 column (GE). Protein fractions were analysed by SDS-PAGE on a 4-12% Bis-Tris gel and protein concentration was measured using a Nanodrop at A280nm.

The *Bi*TLP^FL construct was transformed into *E. coli* LOBSTR cells and selected on LB medium supplemented with 100 µg/µL ampicillin.

Cells were harvested by centrifugation for 30 min at 4000 × g at 4 °C. The bacterial pellets were resuspended in lysis buffer (50 mM Tris-HCl pH 7.5; 500 mM NaCl; 10 mM imidazole; 10% glycerol; 1 mM TCEP-HCl) and lysed using a constant flow cell disruptor. The crude lysate was clarified by centrifugation (27,000 $g$; 30 min; 4C). The clarified lysate was passed through 0.45 and 0.2 µm syringe filters and subsequently applied to a prepacked HisTrapTM HP column (Cytiva). *Bi*TLP^FL was eluted stepwise using wash buffer containing 20 to 500 mM imidazole. Fractions containing *Bi*TLP^FL were concentrated and then loaded onto a Superdex™ 200 Increase 10/300 (Cytiva), and fractions corresponding to were pooled, concentrated to 0.2 mg/ml and stored at −80 °C.

### Cryo-EM sample preparation and data collection
3 µl of purified *Bi*TLP at 0.5 mg/ml was applied onto a plasma cleaned holey carbon grid (Quantifoil® R 1.2/1.3, 300 copper mesh, Agar Scientific; Cat#AGS143-2). The grids were blotted for 3.5 s (blot force=5), at 20 °C with 100% humidity then plunged into liquid nitrogen-cooled liquid ethane using a Vitrobot Mark IV (Thermo Fisher Scientific). Data were collected on a Titan Krios (Thermo Fisher Scientific) equipped with a K3 Summit direct electron detector at a nominal magnification of 105,000x with a calibrated pixel size of 0.83 Å/pixel. Image stacks were acquired with an accumulated dose of 38.3 e⁻/Å² fractionated over 40 frames and a defocus range of −0.8 to −2.2 µm.

For *Bi*TLP^FL, 2 µl of purified sample at 0.2 mg/ml was applied onto a glow discharged holey carbon grid (Quantifoil® R 1.2/1.3, 300 copper mesh). Following a 3 s incubation time, the grid was blotted for 10 s, and then plunged into liquid ethane using the EMGP2 Leica system operating at 4 °C and 95% humidity. Data for *Bi*TLP^FL were collected using CryoARM 300 (JEOL) equipped with an in-column Omega energy filter and K3 direct electron detector (Gatan). Movies of *Bi*TLP^FL were acquired using SerialEM v3.1[99] in super resolution and CDS modes at a nominal magnification of 100,000x corresponding to a calibrated pixel size of 0.2432 Å/pixel. Image stacks were acquired with an accumulated dose of 40 e⁻/Å² fractionated over 40 frames and a defocus range of −0.5 to −2.5 µm.

### Cryo-EM data processing, model building and validation
For *Bi*TLP, preprocessing including motion correction, CTF parameter estimation, particle picking, and extraction was carried out using SIMPLE 3.0[100]. Further processing was carried out in cryoSPARC[101]. Junk particles were discarded through iterative rounds of 2D classification. Five ab initio model were generated. Particles were further 3D classified through heterogenous refinement. The best map was further refined using homogenous refinement. Following particle polishing in RELION[102], particles were re-imported into CryoSPARC[101] and after two rounds of 2D classification, the map was further refined by homogenous refinement with a total of 3,010,268 particles included in the final map reached an average resolution of 2.06 Å based on the FSC = 0.143 criteria as estimated in CryoSPARC[101].

The final map was sharpened by applying a uniform, inverse B-factor of −50 Å², following which an AlphaFold model of *Bi*TLP was docked into the sharpened map, using the 'dock in map' tool in PHENIX[103]. The fitted model was subjected to iterative rounds of manual refinement in COOT[104] and real space refinement in PHENIX[103]. The final model was validated using MolProbity[105] within PHENIX[103].

Data processing for *Bi*TLP^FL was performed in cryoSPARC[101]. Iterative rounds of 2D classification were conducted to remove junk particles. 2D class rebalancing was then performed and the resultant particles used for an ab initio model. Heterogenous refinement was then performed, and the highest quality map was selected for further refinement using homogenous refinement. This map was then used as the reference map for further rounds of 3D classification using exported particles from cryoSPARC[101] to RELION[102]. Improvement of the β-propeller density was observed using RELION[102]. The

corresponding particles and map were re-imported back into cryoS-PARC for homogenous refinement, yielding a final map at 3.74 Å resolution. The map was sharpened using DeepEMhancer to improve map resolvability[106].

An AlphaFold model of *Bi*TLP[FL] was then generated and fitted into the map using the Namdinator server[107]. Manual model building using COOT[104] was then performed and molecular dynamic flexible fitting was performed using ISOLDE (ChimeraX plugin[108];). Real space refinement and validation were then performed in PHENIX[103].

All figures were prepared using UCSF ChimeraX v.1.8[109].

## Western blot

Samples were separated on a 4-12% Bis-Tris NuPAGE gel and transferred to a nitrocellulose membrane by electrophoresis. The membrane was blocked with 3% (w/v) BSA in PBS buffer supplemented with 0.01% (v/v) Tween-20 for 30 min at room temperature. After that, the membrane was sequentially incubated with primary and secondary HRP-conjugated antibodies prior to signal detection with ECL reagents (RPN2106, VWR). Depending on the tag of interest, commercial monoclonal anti-HA (sigma Aldrich, H3663-200UL), anti-strepII (IBA, 2-1507-001) or anti-FLAG (sigma Aldrich, F1804-50UG) antibodies were used at 1:1000 dilution. Secondary antibody anti-mouse IgG HRP (ThermoFisher, 31430) was used at 1:10000 dilution.

## Molecular dynamics simulations

The structure of *Bacillus inaquosorum* CTD used for molecular dynamics simulation was generated using Alphafold2[49]. The protein was mapped to a Coarse-grained representation using Martinize2[110]. To study membrane association, a system containing one protein positioned in the solvent, above a POPG:POPE (3:1) membrane was built. All systems were built using the insane[111] python script. To study membrane deformation, we constructed a system containing four proteins already positioned at proximal the membrane based on the final snapshot of a system containing one protein.

All simulations were performed with GROMACS 2020.5 and the MARTINI3 force field[112]. The Bussi–Donadio–Parrinello (V-rescale)[113] thermostat was used to control the temperature. During equilibration, the Berendsen barostat6 ($\tau$p = 3 ps) was used, while the Parrinello-Rahman barostat[114] ($\tau$p = 12 ps) was used for production runs. In all cases, the barostat was semi-isotropic, the reference pressure was set to 1 bar and the compressibility to $3.0\times10^{-4}$ bar$^{-1}$. Electrostatic interactions and Lennard-Jones interactions were cutoff at 1.1 nm using the reaction-field[115] method and the potential-shift Verlet method. Bonds between beads were constrained to equilibrium values using the LINCS algorithm[116].

After a short minimization of 5000 step, the systems were equilibrated at 300 K in the NPT ensemble for 200 ps, using a 20 fs time-step, with the proteins being harmonically restrained. Production runs were performed then in the NPT ensemble. Ten runs of 3 $\mu$s were performed for the mono-protein system, while a production run of 10 $\mu$s was set up for the 4-protein system.

Distance analysis was performed as previously described in Jackson et al.[117]. Analysis of the membrane deformation was performed using a previous script[118]. Briefly, the script segments the membrane in patches of defined sizes and computes their local norm. The averaged dot product between local norms and the z axis was then used to quantify the deformation of the membrane.

## Propidium iodide staining

Bacterial cells expressing either the empty vector, *Bi*TLP[CTD] or sp-*Bi*TLP[CTD] were incubated with propidium iodide (stock 1 mg/ml) at room temperature for 10 min. Cells were then centrifuged at 5000 *g* for 3 min and pellets are resuspended with fresh M9 media. 5 μl of cell resuspension was applied onto agar pad and air dried under sterile condition prior to sealing with with a coverslip. Cells were imaged using the Oxford Nanoimager-S with 100x/1.49 oil immersion objective lens and a pixel size of 117 nm. Each image was a composite of 200 frames, each with 100 ms exposure, and cells were imaged at an angle of 49°. Images were analysed using the MicrobeJ[119] plugin for ImageJ[120], and plotted using Graphpad (version 9 for MacOS, GraphPad Software, San Diego, California USA, hQps://www.graphpad.com/).

## Toxicity assay

Each CTD construct was transformed into *E. coli* Top10 cells. An overnight LB pre-culture was used to inoculate 20 mL of LB medium, and protein expression was induced when the OD$_{600}$ reached 0.5-0.6 by adding 2% L-arabinose. OD$_{600}$ was measured every hour for 6 hours post-induction. Bacterial cells were then harvested by centrifugation and stored for small affinity pulldown on streptavidin agarose beads. For each construct, an overnight culture was serially diluted, and 3 μL of the culture was spotted onto a soft agar plate supplemented with 2% arabinose and 100 μg/mL ampicillin, then incubated overnight at 37 °C

## NAD glo assay

Relative intracellular NAD+ levels were quantified in cell lysate using the NAD/NADH-glo bioluminescence assay as per the instructions of the manufacturer (Promega). NAD bioluminescence was recorded on a CLARIOstar Plus (BMG Labtech) microplate reader, for 2 h. Statistical analysis using a one-way ANOVA test, with a Tukey's post-hoc test was performed using GraphPad Prism (version 9 for MacOS, GraphPad Software, San Diego, California USA, hQps://www.graphpad.com/).

## Reporting summary

Further information on research design is available in the Nature Portfolio Reporting Summary linked to this article.

## Data availability

The cryo-EM maps have been deposited in the Electron Microscopy Data Bank (EMDB) under accession codes EMD-52847 (*Bi*TLP); and EMD-71831 (*Bi*TLP[FL]). The atomic coordinates have been deposited in the Protein Data Bank (PDB) under accession codes 9IFO (*Bi*TLP); and 9PT5 (*Bi*TLP[FL]). The source data underlying Figs. 2, 3b and e, 4d and Supplementary Fig. 6c are provide as Source Data files. Unedited gels and Western blots are shown in Supplementary Fig. 8. Molecular dynamics trajectories are available from the corresponding authors upon request. Source data are provided with this paper.

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

## Acknowledgements

We acknowledge the Central Oxford Structural Molecular Imaging Center (COSMIC) electron microscopy where the cryo-EM data of *Bi*TLP was collected. We also thank Dr Joseph Caesar for support with data processing. We would also like to acknowledge Australian Microscopy and Microanalysis Research Facility at the Center for Microscopy and Microanalysis located at the University of Queensland where the data for *Bi*TLP^FL is collected. We also thank Professor Steve Kelly for his valuable help and advice on the phylogenetic tree inference. We also give thanks to the Fribourg Lab for supplying the petMCN vector used to clone *Bacillus inaquosorum* TLP, and to Dr. Valentine Lagage and Dr. Diviya Choudhary for providing the pBAD vector used to clone all bacterial TLP CTD and immunity genes. We finally thank the Dean lab for providing us with the Alphafold model of *Bacillus inaquosorum* TLP. F.R. was funded by the Browne research fellowship, the Queen's college Oxford. J.B. is funded by a Wellcome Trust Biomedical Resource Grant (number 218205/Z/19/Z, PubMLST: Disseminating and exploiting bacterial diversity data for public health benefit). Research in the E.S. lab was supported by the Wellcome Trust (202827/Z/16/Z and 226647/Z/22/Z) and the EMBO Young Investigator Programme.

## Author contributions

F.R. and E.S. conceptualized the study and designed the experiments. F.R. and C.M.R. performed the biochemical experiments. J.C.Z. and Y.S.L. performed the structural biology experiments. J.C.Z. and L.B. established and implemented the cryo-EM methodology in the laboratory. F.R., E.S., K.E.O., and E.L. built and refined the *Bi*TLP model, while Y.S.L. and M.J.L. built and refined the *Bi*TLP^FL model. J.B. performed the bioinformatic analyses. J.S. performed the microscopy experiments. A.S. and M.C. performed the structural predictions and molecular dynamics simulations. F.R., J.S., A.S., J.C.Z. and J.B. analysed the data. F.R., J.S., A.S., J.B., C.M.R., Y.S.L., M.J.L., J.S.L., C.K., M.C., M.C.J.M., and E.S. wrote the manuscript.

## Competing interests

The authors declare no competing interests.
