## [Transparent Peer Review file · Nature Communications]

Ancestral neuronal receptors are bacterial accessory toxins

Corresponding Author: Professor Elena Seiradake

Version 0:

Reviewer comments:

Reviewer #1

(Remarks to the Author)

The manuscript "Ancestral neuronal receptors are bacterial accessory toxins." presents a compelling and significant re-evaluation of the evolutionary origins of Teneurins, a family of neuronal receptors crucial in metazoans.

Summary of the paper: The study provides robust evidence that metazoan Teneurins originated from bacterial Teneurin-like proteins (TLPs), fundamentally repurposing an ancient bacterial toxin system for cell-to-cell communication in multicellular organisms. The authors meticulously characterize bacterial TLPs as a novel class of polymorphic toxins, detailing their unique structural features, diverse cytotoxic mechanisms, and the immunity systems co-expressed to protect the host bacteria. Furthermore, the research suggests that bacterial TLPs are integral to complex bacterial social behaviors, encompassing inter-cellular competition and communication within microbial communities.

General assessment: This study offers a profound shift in our understanding of the molecular evolution and the origins of Teneurins. The findings are well-supported by rigorous experimental data, including a high-resolution cryo-EM structure of a TLP and functional assays. The paper is clearly written, and its conclusions are highly impactful.

Major Strengths:

1. **Evolutionary insight:** The paper demonstrates that metazoan Teneurins are repurposed bacterial toxins, acquired through horizontal gene transfer, marking a foundational event in the evolution of metazoans. This redefines the evolutionary trajectory of a key neuronal receptor family.
2. **Detailed structural elucidation:** The cryo-EM structure of *Bacillus inaquosorum* TLP (BiTLP) at 2.1 Å resolution provides unprecedented insights into its molecular architecture. It highlights the conserved superfold homology with metazoan Teneurins while revealing critical differences, such as the cleavage at the N-terminal and C-terminal (absent in metazoans) and the complete encapsulation of the C-terminal toxin domain (CTD) within the Rhs/YD-shell.
3. **Functional characterization of toxins:** The study elucidates the diverse cytotoxic mechanisms of the encapsulated CTDs. It demonstrates that these highly polymorphic CTDs function as potent effector molecules, capable of membrane disruption (e.g., BiTLP CTD) and enzymatic activities like ADP-ribosyl transferase (ART) activity (e.g., *Methylosarcina fibrata* TLP CTD), leading to NAD⁺ depletion and growth inhibition in target cells.
4. **Discovery and validation of immunity systems:** Identification and functional validation of cognate immunity genes co-expressed with TLP-encoding operons. Experimental evidence, including coimmunoprecipitation and functional assays, confirms that these immunity proteins directly bind to and neutralize their corresponding C-terminal toxins, providing a sophisticated self-protection mechanism for the host bacteria.

Major revisions:

Observing a chain break within the FN-plug domain of BiTLP (between residues E722 and S724) is interesting and points towards an evolutionary conservation between bacterial TLPs and Rhs toxins. Still, the manuscript provides only limited information regarding the nature and significance of this N-terminal cleavage. The authors do not identify a conserved motif, provide biochemical validation of the cleavage site (e.g., N-terminal sequencing by Edman degradation), or explore whether this cleavage is conserved across TLPs. Clarifying whether this event is enzymatic, functionally required, or a structural consequence would strengthen the structural and mechanistic conclusions. Additional data or discussion on this point would

be valuable.

The authors identify a conserved cleavage site in the BiTLP C-terminal region and suggest that an aspartyl protease catalytic triad (R2111, D2116, D2140) mediates autoproteolysis, in analogy to RHS toxins. However, canonical RHS toxins typically use a bipartite DPxG-X₁₈-DPxG motif, with both aspartates involved in catalysis. It would strengthen the manuscript to clarify whether this whole motif is present and conserved in BiTLP and related TLPs. A more explicit alignment or structural comparison with the characterized RHS aspartyl protease motif would help place the TLP cleavage mechanism in its proper evolutionary and functional context.

Additionally, to my knowledge, the C-terminal autoproteolytic cleavage in Rhs toxins does not involve an arginine residue. Only the catalytic dyad comprising the two aspartic residues within the motif is believed to participate in catalysis. Could the authors experimentally demonstrate, through point mutations, whether TLPs utilize the proposed catalytic triad for C-terminal autoproteolytic cleavage?

While the authors infer that bacterial TLPs are surface-localized—based on signal peptide predictions and the presence of cell wall/membrane-binding domains—this conclusion remains speculative in the absence of direct experimental evidence. The paper would benefit from validating the subcellular localization of at least one representative TLP, for example, via surface immunostaining, protease accessibility assays, or surface biotinylation. This is particularly relevant given the functional model that TLPs mediate intercellular interactions or deliver toxins at the cell surface, and that the authors propose TLPs are the direct evolutionary ancestors of Teneurins; therefore, experimentally demonstrating that TLPs are surface-exposed would, in my opinion, strengthen their claim.

Reviewer #2

(Remarks to the Author)

Raoelijaona et al. use a mammalian teneurin as a bait to identify teneurin-like proteins (TLP) in bacteria, solve the crystal structure of a representative TLP, and demonstrate TLP-mediated bacterial toxicity. Based on the existing literature, which is referred to by the authors in the introduction and discussion, the presented findings were not fully unexpected to me. However, the large body of work and complementary methodologies provide in-depth knowledge on TLPs that will advance the field of bacterial toxins and shed light on the related use of conserved protein domains among prokaryotic and eukaryotic organisms. I expect that the presented work will be of broad interest to the readers of Nature Communications. In my opinion, the experiments have been done thoroughly, the data is presented clearly, and the writing is excellent.

Few minor comments

1. Abstract: I am afraid that I find some of the statements in here a bit strong. "We show that bacterial Teneurins are accessory proteins, that harbour diverse encapsulated toxins." I do not understand the meaning of "accessory" here.

"They predominate in species with complex social behaviours, suggesting roles in both cell-to-cell competition and cooperation." Is there any other indication for a role in "cooperation" other than the mere presence in these species?

"Their acquisition was a foundational event in the evolution of metazoans." I might have missed this, how do the authors know that this "was" a "foundational" event?

2. I am afraid there is one part that I don't fully get. In the introduction, the authors state that "Phylogenetic analyses suggest Teneurin genes arose following the fusion of a prokaryotic proteinaceous toxin containing a RHS/YD-repeat domain with a eukaryotic gene 1,42,43." After doing their analysis, do the authors still agree with this previous observation? If the authors found teneurin-like proteins in bacteria, does this mean that there was no fusion necessary during the transition from prokaryotes to complex organisms?

3. "In Gram-negative bacteria, particularly within the Myxococcota phylum, TLPs utilize the SEC/SPII secretion system." Is this based on identified signal sequences and therefore predicted? Or has this been experimentally shown? I suggest to clarify.

Version 1:

Reviewer comments:

Reviewer #1

(Remarks to the Author)

I have reviewed the revised manuscript NCOMMS-25-24782A, "Ancestral neuronal receptors are bacterial accessory toxins." The authors have satisfactorily addressed my initial concerns, providing additional experimental results and clarifying their conclusions when based on bioinformatic analysis. Taking all of this into account, I am happy to support the

acceptance for publication of the manuscript in Nature Communications.

Reviewer #2

(Remarks to the Author)

I thank the authors for their additional work. My comments have been addressed to my satisfaction.

Authors: We thank the reviewers for their constructive comments which have helped us to further improve the manuscript by doing additional experiments and revising the draft.

REVIEWER COMMENTS

Reviewer #1 (Remarks to the Author):

The manuscript "Ancestral neuronal receptors are bacterial accessory toxins." presents a compelling and significant re-evaluation of the evolutionary origins of Teneurins, a family of neuronal receptors crucial in metazoans.

Summary of the paper: The study provides robust evidence that metazoan Teneurins originated from bacterial Teneurin-like proteins (TLPs), fundamentally repurposing an ancient bacterial toxin system for cell-to-cell communication in multicellular organisms. The authors meticulously characterize bacterial TLPs as a novel class of polymorphic toxins, detailing their unique structural features, diverse cytotoxic mechanisms, and the immunity systems co-expressed to protect the host bacteria. Furthermore, the research suggests that bacterial TLPs are integral to complex bacterial social behaviors, encompassing inter-cellular competition and communication within microbial communities.

General assessment: This study offers a profound shift in our understanding of the molecular evolution and the origins of Teneurins. The findings are well-supported by rigorous experimental data, including a high-resolution cryo-EM structure of a TLP and functional assays. The paper is clearly written, and its conclusions are highly impactful.

Major Strengths:

1. **Evolutionary insight:** The paper demonstrates that metazoan Teneurins are repurposed bacterial toxins, acquired through horizontal gene transfer, marking a foundational event in the evolution of metazoans. This redefines the evolutionary trajectory of a key neuronal receptor family.
2. **Detailed structural elucidation:** The cryo-EM structure of *Bacillus inaquosorum* TLP (BiTLP) at 2.1 Å resolution provides unprecedented insights into its molecular architecture. It highlights the conserved superfold homology with metazoan Teneurins while revealing critical differences, such as the cleavage at the N-terminal and C-terminal (absent in metazoans) and the complete encapsulation of the C-terminal toxin domain (CTD) within the Rhs/YD-shell.
3. **Functional characterization of toxins:** The study elucidates the diverse cytotoxic mechanisms of the encapsulated CTDs. It demonstrates that these highly polymorphic CTDs function as potent effector molecules, capable of membrane disruption (e.g., BiTLP CTD) and enzymatic activities like ADP-ribosyl transferase (ART) activity (e.g., *Methylosarcina fibrata* TLP CTD), leading to NAD⁺ depletion and growth inhibition in target cells.
4. **Discovery and validation of immunity systems:** Identification and functional validation of cognate immunity genes co-expressed with TLP-encoding operons. Experimental evidence, including coimmunoprecipitation and functional assays, confirms that these immunity proteins directly bind to and neutralize their corresponding C-terminal toxins, providing a sophisticated self-protection mechanism for the host bacteria.

Authors: We thank the reviewers for their positive assessment and comprehensive summary.

Major revisions:

Observing a chain break within the FN-plug domain of BiTLP (between residues E722 and S724) is interesting and points towards an evolutionary conservation between bacterial TLPs and Rhs toxins. Still, the manuscript provides only limited information regarding the nature and significance of this N-terminal cleavage. The authors do not identify a conserved motif, provide biochemical validation of the cleavage site (e.g., N-terminal sequencing by Edman degradation), or explore whether this cleavage is conserved across TLPs. Clarifying whether this event is enzymatic, functionally required, or a structural consequence would strengthen the structural and mechanistic conclusions. Additional data or discussion on this point would be valuable.

Authors: We agree these are interesting questions, and in response to this comment, we have carried out the following additional experiments.

We performed N-terminal sequencing, which confirmed that cleavage occurs immediately upstream S724 (included separately, as supplementary data). Furthermore, we have included in our revised manuscript a second cryo-EM structure of BiTLP, obtained from the full-length construct which was purified under reducing conditions (included as supplementary Fig. 4). The structure is almost identical to our initially reported structure but is more clearly resolved in some regions (despite the inferior overall resolution) and shows a clear and unequivocal absence of density prior to Ser724 (included as supplementary Fig. 5, panel d) entirely consistent with the N-terminal cleavage described for our first structure and supported by the N-terminal sequencing (supplementary data). The density also stops abruptly beyond Leu2144, consistent with the reported aspartic protease autoproteolysis site. This new cryo-EM map and the N-terminal sequencing analyses therefore provide conclusive structural and biochemical evidence supporting cleavage N-terminal to Ser724 and strengthening the evidence for an evolutionary conservation between bacterial TLPs and RHS toxins.

Elucidation of further details concerning the mechanism of proteolysis is challenging, as for T6SS-associated RHS effectors there does not appear to be a conserved mechanism of proteolysis. In the case of *Acinetobacter Baumannii* Tse15¹, cleavage is observed N-terminal to a moderately conserved serine and is inhibited by alanine substitution of a proposed nucleophilic glutamate located 7 residues downstream (but not by mutation of the serine). Most other RHS effectors studied to date describe N-terminal cleavage at or near a conserved proline residue, and while many consensus motifs have been proposed, there appears to be no pattern or conservation of obvious catalytic residues amongst these. Taken together, defining a conserved motif for the cleavage is rather challenging as reflected by other T6SS-associated RHS effectors. Nevertheless, the serine residue is highly conserved across all TLP (included as supplementary Fig. 3 panel f) sequences, suggesting that the n-terminal cleavage mechanism could be conserved among TLP family members. Furthermore, autocatalytic cleavage upstream of the shell domain has been observed in other T6SS-associated RHS effectors¹⁻⁵.

Interestingly, while the beta-propeller domain is also more clearly resolved in our second map, the FN-plug domain is not well resolved, and there is minimal to no resolvable density attributable to the CTD within the YD shell, consistent with a high degree of conformational flexibility associated with these domains. It is tempting to suggest that this second structure may represent a progressed conformational state where the FN plug and CTD have started to emerge from the cage post-cleavage, which might in turn suggest that the additional ~300 residue adhesin domain included in this construct might assist with release, but equally this

could be a consequence of the reducing conditions under which the experiment was performed. We therefore decided not to make any further claims in relation to this in the current manuscript and purely present the structure as evidence of cleavage.

The authors identify a conserved cleavage site in the BiTLP C-terminal region and suggest that an aspartyl protease catalytic triad (R2111, D2116, D2140) mediates autoproteolysis, in analogy to RHS toxins. However, canonical RHS toxins typically use a bipartite DPxG–X₁₈–DPxG motif, with both aspartates involved in catalysis. It would strengthen the manuscript to clarify whether this whole motif is present and conserved in BiTLP and related TLPs. A more explicit alignment or structural comparison with the characterized RHS aspartyl protease motif would help place the TLP cleavage mechanism in its proper evolutionary and functional context.

Authors: As suggested, we now provide a short sequence alignment comparing the BiTLP sequence with consensus cleavage motif sequences, in supplementary Fig. 5a. In TLPs, the linker between the two bipartite DPxG motifs contains two additional residues, and the glycine residue of the first DPxG is sometimes substituted by a leucine or an arginine. We are pointing this out in the revised results section, : *“In analogy to the bipartite DPxG-X₁₈-DPxG motif found in other RHS proteins, where the two aspartic acids both play a role in catalysis, we found two DPxG motives twenty residues apart from each other. In some species, the glycine residue for the first motif is substituted by a leucine or an arginine (supplementary Fig. 5a-c). The structure of the catalytic aspartic acid motif is conserved and is located proximal to a structurally conserved arginine which has previously been shown as essential for catalysis (Fig. 2d, supplementary Fig. 5b). Indeed, mutation of this arginine reduces autoproteolytic activity (supplementary Fig. 5a-c).”*

Additionally, to my knowledge, the C-terminal autoproteolytic cleavage in Rhs toxins does not involve an arginine residue. Only the catalytic dyad comprising the two aspartic residues within the motif is believed to participate in catalysis. Could the authors experimentally demonstrate, through point mutations, whether TLPs utilize the proposed catalytic triad for C-terminal autoproteolytic cleavage?

Authors: We are particularly grateful for this comment because it (i) drew our attention to a mistake in our annotation of residue numbers; and (ii) suggested additional experiments which we have now completed. The conserved arginine we refer to is at position 2103 rather than 2111 (supplementary Fig. 5a) and indeed the importance of this arginine in catalysis has already been established by some members of our team in the original study of RHS proteins (Busby et al. 2012). To validate the importance of this residue, we expressed a construct in which arginine 2103 is mutated to alanine and analysed cleavage products using Coomassie staining and western blotting. This showed that autoproteolytic cleavage is indeed reduced in the mutant, albeit not completely abolished (supplementary Fig. 5c). We have corrected the numbering error in our manuscript and added the following statement to our revised manuscript, describing these results *“Indeed, mutation of this arginine reduces autoproteolytic activity (supplementary Fig. 5a-c), consistent with what has been established previously for the RHS subunit of bacterial ABC toxins⁶.”*

Interestingly, the proteolytic cleavage we describe upstream of the YD shell, in the FN-plug domain, is affected by the mutation. It is possible that correct localisation of the cleaved CTD, which is located in proximity to the FN-plug in our structure, is important to support cleavage in this upstream, distinct area of the protein.

While the authors infer that bacterial TLPs are surface-localized—based on signal peptide predictions and the presence of cell wall/membrane-binding domains—this conclusion remains speculative in the absence of direct experimental evidence. The paper would benefit from validating the subcellular localization of at least one representative TLP, for example, via surface immunostaining, protease accessibility assays, or surface biotinylation. This is particularly relevant given the functional model that TLPs mediate intercellular interactions or deliver toxins at the cell surface, and that the authors propose TLPs are the direct evolutionary ancestors of Teneurins; therefore, experimentally demonstrating that TLPs are surface-exposed would, in my opinion, strengthen their claim.

Authors: We are very grateful for the suggestion and acknowledge that the evidence for cell surface localisation presented in our current manuscript is limited to bioinformatical analysis. We agree that including additional experimental evidence supporting this would significantly strengthen our claims that TLPs may be surface localised, and have indeed attempted to carry out additional experiments, as suggested by the reviewer, that would substantiate this. Unfortunately, despite our success in generating host strains carrying labelled *Bi*TLP genes for visualisation, we are yet to find a strain where the protein is successfully expressed, perhaps due to a lack of necessary protein folding machineries, or possibly pointing to an essential role of other, as yet uncharacterised accessory genes.

Experimental procedure to investigate *Bi*TLP localisation in *Bacillus subtilis*

a. *B. subtilis* growth on soft agar using chloramphenicol for selection of strain carrying *Bi*TLP gene.

b. Western blot analysis to assess protein expression. *B. subtilis* cells were grown in LB at 37°C, induced with 0.5% xylose, and incubated overnight at 20°C. As a FLAG tag was introduced into the *Bi*TLP sequence, we used an anti-FLAG antibody to detect protein expression. A purified protein sample was included as a positive control. Both soluble (S) and membrane (M) fractions were analysed to determine protein localisation, but we could not find any evidence for *Bi*TLP expression in *B. subtilis*.

c. Anti-FLAG immunostaining of *B. subtilis* cells showing that, compared to the wild-type (negative control), the modified strain is labelled equally by the antibody. This experiment also suggests no expression of *Bi*TLP in the modified *B. subtilis* strain.

Despite these challenges, we do feel that unequivocally proving TLPs to be cell surface localised is not essential to the main theme of this manuscript, which identifies TLPs as close structural homologues of eukaryotic teneurins that share functional and mechanistic features with bacterial proteins that function primarily as toxins, establishing evolutionary relationships between these two classes of proteins and how teneurins were ultimately co-opted. While the bioinformatic analyses provide guiding evidence for an additional compelling hypothesis regarding cellular localization of TLPs, we acknowledge that such conclusions remain somewhat speculative. We have therefore modified our manuscript in the results section (lines 102-107) acknowledging that these are predicted or putative functions, and in the discussion (starting on line 298) as follows: “*While the evidence we present here for cell surface localization of TLPs is limited to functional homologies inferred from bioinformatic sequence analyses, the predicted cell surface localization of bacterial TLPs to the cell surface may also allow them to engage in cell-to-cell contact and signalling functions.*”

Reviewer #2 (Remarks to the Author):

Raelijaona et al. use a mammalian teneurin as a bait to identify teneurin-like proteins (TLP) in bacteria, solve the crystal structure of a representative TLP, and demonstrate TLP-mediated bacterial toxicity. Based on the existing literature, which is referred to by the authors in the introduction and discussion, the presented findings were not fully unexpected to me. However, the large body of work and complementary methodologies provide in-depth knowledge on TLPs that will advance the field of bacterial toxins and shed light on the related use of conserved protein domains among prokaryotic and eukaryotic organisms. I expect that the presented work will be of broad interest to the readers of Nature Communications. In my opinion, the experiments have been done thoroughly, the data is presented clearly, and the writing is excellent.

Authors: thank you for the careful evaluation and positive comments.

Few minor comments

1. Abstract: I am afraid that I find some of the statements in here a bit strong.

“We show that bacterial Teneurins are accessory proteins, that harbour diverse encapsulated toxins.” I do not understand the meaning of “accessory” here.

Authors: the logic behind using the word “accessory” is that, although TLP proteins are found across the bacterial kingdom (i.e. in different phyla, gram positive and gram negative), our analysis revealed that only a small subset of the investigated species encode this gene. This suggests that TLPs are not essential for basic bacterial survival but seem to be selected for in specific environments or under specific circumstances. In contrast, Teneurins are present in every single higher metazoan species with a centralised nervous system, both vertebrate and invertebrate. We have modified a sentence in the abstract, to try to make the meaning of “accessory” in this context more clear” *They are rare but widely distributed across bacterial taxa, and are predominantly seen in species with complex social behaviours, suggesting roles in cell-to-cell interaction.*”

“They predominate in species with complex social behaviours, suggesting roles in both cell-to-cell competition and cooperation.” Is there any other indication for a role in “cooperation” other than the mere presence in these species?

Authors: indeed, most published studies suggest that RHS proteins play roles in bacterial competition. However, in *Myxococcus xanthus*, a RHS homolog has been reported to be involved in Social motility ⁷ We have softened and shortened our statement in the abstract: “[TLPs] are predominantly seen in species with complex social behaviours, suggesting roles in cell-to-cell interaction.”

“Their acquisition was a foundational event in the evolution of metazoans.” I might have missed this; how do the authors know that this “was” a “foundational” event?

Authors: thanks for raising this, it is a valid point. Overall, it is inherently difficult to prove whether the acquisition of a gene millions of years ago was a necessary (or ‘foundational’) event for the evolution of a certain trait. We infer this from the observation that Teneurins are essential in all higher metazoan organisms (e.g. they are always present and knocking them out in species that have only one copy of Teneurin, such as *C. elegans*, leads to major defects during development), and that they don’t seem to be essential in most bacteria. But we agree that this is not strictly a proof and have tweaked our abstract to end in: “...*This work confirms that metazoan Teneurins are repurposed bacterial toxins that have evolved to be essential mediators of intercellular communication in all advanced nervous systems. Their acquisition was a key event in the evolution of metazoans.*”

2. I am afraid there is one part that I don’t fully get. In the introduction, the authors state that “Phylogenetic analyses suggest Teneurin genes arose following the fusion of a prokaryotic proteinaceous toxin containing a RHS/YD-repeat domain with a eukaryotic gene 1,42,43.” After doing their analysis, do the authors still agree with this previous observation? If the authors found teneurin-like proteins in bacteria, does this mean that there was no fusion necessary during the transition from prokaryotes to complex organisms?

Authors: Eukaryotic Teneurins and the bacterial TLPs share the presence of a conserved “Teneurin superfold” (FN-plug, NHL, and YD-shell) plus the upstream TTR and cysteine rich domains. All of these were most likely acquired by eukaryotes “in one piece” by the proposed HGT event, and this is still supported by our new data. The domains upstream of these sequences are divergent in bacteria, but they are largely conserved in the eukaryotic Teneurins

which contain a transmembrane helix and string of epidermal growth factor-like (EGF) domains. These are similar to EGF domains found in the eukaryotic extracellular matrix protein ‘Tenascin’, from which Teneurin takes its name^{8,9}. It this chimeric arrangement (N-terminal ‘eukaryotic’ EGFs and C-terminal ‘bacterial’ toxin-like domains) that gave rise to the notion that Teneurins are derived from a gene fusion event¹⁰. To better explain this, we now amended the following sentence in the introduction, to read: “*Phylogenetic analyses suggest Teneurin genes arose following the fusion of a prokaryotic proteinaceous toxin containing a RHS/YD-repeat domain (the Teneurin ‘superfold’ and flanking domains) with a eukaryotic transmembrane protein gene.*”

3. “In Gram-negative bacteria, particularly within the Myxococcota phylum, TLPs utilize the SEC/SPII secretion system.” Is this based on identified signal sequences and therefore predicted? Or has this been experimentally shown? I suggest to clarify.

Authors: yes indeed, this is based on predicted signal sequences. We have amended the following sentence in line 100-106: “*Based on putative signal sequences, the Gram-negative TLPs, such as those within the Myxococcota phylum, are predicted to be targeted to the SEC/SPII secretion system. These TLPs also contain predicted lipoprotein domains that could facilitate attachment to the membrane surface. In contrast, putative signal sequences found in TLPs from Gram-positive bacteria are predicted to use the SEC/SPI secretion system and contain predicted S-layer or cell wall binding domains that would anchor them to the cell surface*”

References:

1. Hayes, B. K. *et al.* Structure of a Rhs effector clade domain provides mechanistic insights into type VI secretion system toxin delivery. *Nat Commun* **15**, 8709 (2024).
2. Pei, T.-T. *et al.* Intramolecular chaperone-mediated secretion of an Rhs effector toxin by a type VI secretion system. *Nat Commun* **11**, 1865 (2020).
3. Jurėnas, D. *et al.* Mounting, structure and autocleavage of a type VI secretion-associated Rhs polymorphic toxin. *Nat Commun* **12**, 6998 (2021).
4. Günther, P. *et al.* Structure of a bacterial Rhs effector exported by the type VI secretion system. *PLOS Pathogens* **18**, e1010182 (2022).
5. González-Magaña, A. *et al.* Structural and functional insights into the delivery of a bacterial Rhs pore-forming toxin to the membrane. *Nat Commun* **14**, 7808 (2023).

6. Busby, J. N., Panjikar, S., Landsberg, M. J., Hurst, M. R. H. & Lott, J. S. The BC component of ABC toxins is an RHS-repeat-containing protein encapsulation device. *Nature* **501**, 547–550 (2013).
7. Youderian, P. & Hartzell, P. L. Triple Mutants Uncover Three New Genes Required for Social Motility in *Myxococcus xanthus*. *Genetics* **177**, 557–566 (2007).
8. Tucker, R. P. & Chiquet-Ehrismann, R. Evidence for the evolution of tenascin and fibronectin early in the chordate lineage. *The International Journal of Biochemistry & Cell Biology* **41**, 424–434 (2009).
9. Wides, R. The Natural History of Teneurins: A Billion Years of Evolution in Three Key Steps. *Front. Neurosci.* **13**, (2019).
10. Tucker, R. P., Beckmann, J., Leachman, N. T., Schöler, J. & Chiquet-Ehrismann, R. Phylogenetic Analysis of the Teneurins: Conserved Features and Premetazoan Ancestry. *Mol Biol Evol* **29**, 1019–1029 (2012).